# Modulations of Photosynthetic Membrane Lipids and Fatty Acids in Response to High Light in Brown Algae (*Undaria pinnatifida*)

**DOI:** 10.3390/plants14121818

**Published:** 2025-06-13

**Authors:** Natalia V. Zhukova, Irina M. Yakovleva

**Affiliations:** National Scientific Center of Marine Biology, Far Eastern Branch, Russian Academy of Sciences, Vladivostok 690041, Russia; yakovleva72@mail.ru

**Keywords:** brown macrophytes, chloroplast membrane, monogalactosyldiacylglycerol, phosphatidylglycerol, fatty acids, high light, photosynthesis

## Abstract

Light is a source of energy for photosynthesis and hence promotes the regulation of multiple physiological and metabolic processes in photoautotrophic organisms. Understanding how brown macrophytes adjust the physical and biochemical properties of photosynthetic membranes in response to high-irradiance environments has received little attention so far. Particularly, it concerns the lipid flexibility of thylakoid membranes. We examined the lipid classes, fatty acid (FA) profiles, chloroplast ultrastructure, and photosynthetic performance of the brown macroalga *Undaria pinnatifida* after long-term exposure to high light (HL) and moderate light (ML) intensities, at 400 and 270 µmol photons m^−2^ s^−1^, respectively. *U. pinnatifida* responded to HL with a reduction in the level of thylakoid membrane lipids, monogalactosyldiacylglycerol (MGDG), digalactosyldiacylglycerol (DGDG), sulfoquinovosyldiacylglycerol (SQDG), and phosphatidylglycerol (PG), while the character of lipid modulations was specific. The content of storage lipids, triacylglycerols enriched in n-3 polyunsaturated fatty acids (PUFAs), increased under HL. The general response to long-term HL for the studied thylakoid membrane lipids, but not for SQDG, was the remodeling of FA composition towards increasing the percentages of saturated and monounsaturated acyl groups over PUFAs, suggesting a photoprotective strategy against the intensification of lipid peroxidation. In all, we showed that remodeling in photosynthetic membrane lipids accompanied by structural changes in chloroplasts and modulations in photosynthetic performance augmented the ability of *U. pinnatifida* to counteract high-intensity light, thereby contributing to its survival potential under suboptimal irradiance conditions.

## 1. Introduction

Light is a source of energy for photosynthesis and hence promotes the regulation of multiple physiological and metabolic processes in photoautotrophic organisms, including growth, development, and biomass production [1]. In natural habitats, light intensity fluctuates across short-term (seconds to hours) or long-term (seasonal or locational variation) time scales. Furthermore, the quantity of photosynthetically active radiation (PAR) is frequently in excess of that needed to saturate photosynthesis, particularly during summer periods. The ability of algae and higher plants to adjust photosynthetic properties to these conditions plays an essential role for their survival and metabolic competence [2,3].

Thylakoid membranes of chloroplasts serve a key function in maintaining the stability of photosynthesis. These thylakoid membranes are the sites of photochemical reactions and electron transport chains and contain photosystems (PSs) I and II and their subcomplexes, such as light harvesting complexes D1 and D2 (reaction center proteins of PSII), whose operation directly depends on membrane properties [4]. The fluidity and stability of thylakoid membranes, which are critical for the performance of cells under high-intensity light, are clearly affected by variations in irradiance intensity [5,6,7].

The physico-chemical properties and functioning of thylakoid membranes are determined by their lipid composition [4], which is further essential for the organization of light harvesting complexes and the photosynthetic activity of photoautotrophs [5,8]. Particularly, integral membrane proteins are exposed to a complex and dynamic lipid environment modulated by non-bilayer lipids that can influence protein functions by lipid–protein interactions [9,10]. In eukaryotic chloroplasts, the lipid bilayer of thylakoid membranes is composed mainly of four unique lipids, including monogalactosyldiacylglycerol (MGDG), digalactosyldiacylglycerol (DGDG), sulfoquinovosyldiacylglycerol (SQDG), and PG (phosphatidylglycerol), which are not seen in large quantities in membranes other than thylakoid membranes [8]. These lipids are essential for chloroplast development and thylakoid membrane formation and participate in shaping thylakoid architecture (including membrane packing), electron transfer, and the regulation of photosynthetic activity [8,9,11]. MGDG and DGDG are uncharged glycolipids that make up the main bulk of thylakoid membrane lipids and form a lipid matrix as the main component for photosynthetic complexes. Glycolipid SQDG and phospholipid PG are anionic lipids with negatively charged head groups. The majority of PG molecules function as the site of oxygenic electron transport in PSII [8]. In addition to the composition and ratio of individual lipid classes, the degree of membrane lipid unsaturation is known to be essential for the maintenance of photosynthetic capacity, chlorophyll content, and photosynthetic complexes [2,12,13].

Recent studies have demonstrated the importance of lipid remodeling as an effective response strategy for higher plants to resist the excessive irradiance environment [6,7,14,15]. Some results from these studies reveal that high light intensities induce the redistribution of lipids as well as changes in the extent of acyl group unsaturation within individual lipids of thylakoid and cell membranes, altering their membrane fluidity and permeability. The reorganization of the membrane lipid environment helps prevent membrane lipid damage, supports proper membrane functioning, and contributes to the preservation of the normal structure of the lipid bilayer.

The effects of high light on algal membrane lipids and their fatty acids have been documented mainly for marine and freshwater microalgae [16,17,18,19] and partially for some species of the green and the red macroalgae [13,20,21,22,23]. However, in most of the studies, information is limited to fatty acid remodeling, and/or the duration of the used experimental approaches did not exceed a few hours or days, representing the short-term acclimation response of the species. Furthermore, for brown macroalgae (Phaeophyceae) belonging to the Ochrophyta phylum, information describing the modulations in membrane lipids and their fatty acids in response to high light is still lacking. Meanwhile, the ability of brown macroalgae to alter lipid metabolism in response to environmental influences, particularly irradiance fluctuations, is a critical factor with respect to their growth and survival in a wide range of conditions and also has relevance to the commercial exploitation of macroalgae for the production of lipid-based compounds.

Here, we hypothesize that prolonged exposure to high irradiance, which is near the upper limit of tolerance for the Phaeophyceae species, will induce specific modulations in membrane lipids and their fatty acids, and this remodeling will promote the functional activity of the membranes that will in turn provide an adjustment to the photosynthesis of the algae at high light. To test this hypothesis, we examined adjustments in thylakoid membrane functionality and the flexibility of lipid metabolism in the brown macroalga *Undaria pinnatifida* after long-term exposure to high light. Previously, this species has been used as a model to study mechanisms of low-light acclimation, typical for brown algae [24]. The kelp *U. pinnatifida* (Harv.) Suringar (1873) (Laminariales, Phaeophyceae, Ochrophyta) is an invasive annual macroalga endemic to Asian waters. This species has the ability to successfully colonize various coastal areas, including low-intertidal zones [25] and, thus, may experience prolonged periods of high light exposure, especially in spring after ice melting and during the summer time [26,27]. Additionally, chlorophyll fluorescence, net photosynthesis, dark respiration, and the level of lipid peroxidation were assessed as the proxies for photosynthetic performance, cellular energy requirements, and indices of the oxidative response of the algae, respectively, following high light exposure.

This is the first study to demonstrate the strategy of brown macroalgae to counteract high-intensity light from the view point of lipid metabolism. The obtained data will provide new insights into the responses of the Phaeophyceae species to fluctuating irradiance conditions from a lipidomic perspective.

## 2. Results

### 2.1. Lipid Content and Lipid Class Profiles

The total lipid (TL) contents of *Undaria pinnatifida*, exposed at 270 µmol photons m^−2^ s^−1^ (moderate light, ML) within 35 days, averaged 5.05 ± 0.22 mg·g^−1^ ww, while those of algal samples exposed at 400 µmol photons m^−2^ s^−1^ (high light, HL) averaged 4.07 ± 0.17 mg·g^−1^ ww. The TL content of *U. pinnatifida* was significantly affected by irradiance intensity (ANOVA, *p* < 0.05), showing a 20% reduction under HL.

The major lipid classes of *U. pinnatifida* were presented by glycoglycerolipids (GLs)—monogalactosyldiacylglycerol (MGDG), digalactosyldiacylglycerol (DGDG) and sulfoquinovosyldiacylglycerol (SQDG); phosphoglycerolipids (PLs)—phosphatidylglycerol (PG), phosphatidylcholine (PC), phosphatidylethanolamine (PE), and phosphatidylinositols (PIs); and a neutral (storage) lipid—triacylglycerol (TAG). GLs and PGs are components of thylakoid membranes of chloroplasts, while PC, PE, and PIs are structural components of cell membranes, such as plasma membranes, mitochondrial, etc. Under optimal light growth conditions (ML treatment), glycoglycerolipids were the most representative lipids of chloroplast membranes, accounting for between 42 and 53% of total lipids, with prevalence of MGDG over DGDG and SQDG. Concerning phospholipids, PG, PC, and PE accounted for 10.2–11.4%, 11.4–17.6%, and 10.4–11.4%, respectively, while PIs constituted the less representative lipid class of cell membranes in these experiments. The storage lipid TAG accounted for 11.8–14.4% of total lipids.

Long-term exposure to HL induced significant alterations in the content of the major lipid classes in *U. pinnatifida* (Figure 1). All thylakoid membrane lipids showed pronounced declines in their content, where MGDG, DGDG, SQDG, and PG were reduced by 40%, 28%, 34%, and 33%, respectively, compared to the ML treatment group (Tukey HSD, *p* < 0.05). The contents of cell membrane lipids PC and PE were also reduced by 20% after HL exposure compared to the ML group (Tukey HSD, *p* < 0.05). As for PI, there were no significant differences (Tukey HSD, *p* > 0.05) in the content of this lipid between the light treatments after 35 days experimental period. The content of the neutral storage lipid TAG showed a 1.5-fold increase after HL exposure compared to the ML treatment group (Tukey HSD, *p* < 0.05).

The MGDG-to-DGDG ratio showed a significant reduction in HL-exposed samples compared to the ML treatment group (Tukey HSD, *p* < 0.05; Figure 2). The MGDG-to-Chl and DGDG-to-Chl ratios exhibited higher values (1.5-fold and 1.8-fold, respectively; Tukey HSD, *p* < 0.05) in the HL algae than in the ML algae.

### 2.2. Fatty Acid Profiles

In addition to changes in lipid class composition, HL exposure induced significant alterations in the fatty acid composition of major membrane lipids in *U. pinnatifida* after the 35-day experimental period (Figure 3, Table 1). In MGDG, DGDG, and PG, the content of saturated fatty acids (SFAs) significantly increased with HL (Figure 3). This was mainly caused by the 1.2-1.9-fold increase in the values of 16:0 in MGDG and PG and by the 1.5-fold increase in values of 18:0 in DGDG compared to the ML treatment group (Tukey HSD, *p* < 0.05). The content of monounsaturated fatty acids (MUFAs) in MGDG, DGDG, and PG also varied between the HL and ML treatments. In these lipids, the content of 18:1n-9 exhibited significant increases (1.3-1.4-fold) in HL samples. Meanwhile, the content of 16:1n-3trans in PG showed a pronounced reduction (by a factor of 1.8; Tukey HSD, *p* < 0.05) in HL-exposed algae compared to the ML treatment group. The lipids MGDG, DGDG, and PG had the lowest contents of n-3 PUFAs in HL algae due to reductions in 18:4n-3 and 20:5n-3 in MGDG (by factors of 1.2 and 1.4, respectively), 20:5n-3 in DGDG (by a factor of 1.1), and 18:3n-3 in PG (by a factor of 1.4) compared to the ML group (Tukey HSD, *p* < 0.05). In MGDG, DGDG, and PG, the majority of n-6 PUFAs did not show significant differences (Tukey HSD, *p* > 0.05) in their contents between light treatments. Meanwhile, the contents of 18:2n-6 in MGDG, 18:3n-6 in DGDG, and 18:2n-6 and 20:4n-6 in PG showed a pronounced increase (1.2-2.7-fold; Tukey HSD, *p* < 0.05) in the HL samples compared to ML samples. In SQDG, the fatty acid composition was mostly unchanged in HL samples, except the reduced content of 16:0 (by a factor of 1.1) and increased content of 18:2n-6 (by a factor of 1.5) in the HL treatment compared to the ML treatment (Tukey HSD, *p* < 0.05).

In PC, PE, and PI, there were no significant differences (Tukey HSD, *p* > 0.05) in the contents of the majority of fatty acids between light treatments, except for 16:0 and 20:3n-6 in PC and 18:1n-9, 20:4n-6, and 20:5n-3 in PE (Table 1). In PC, the content of 16:0 was higher while that of 20:3n-6 was lower (by a factor of 1.6) in the HL samples than in the ML treatment (Tukey HSD, *p* < 0.05). In PE, the content of 18:1n-9 increased by a factor of 1.5, while that of 20:4n-6 and 20:5n-3 was reduced by factors of 1.1 and 1.4, respectively, in the HL treatment (Tukey HSD, *p* < 0.05).

In TAG, the contents of the main SFAs, 16:0 and 18:0, were 1.4 and 1.7-fold lower in the HL samples than that in the ML treatment group (Tukey HSD, *p* < 0.05, Table 1). The contents of the majority of MUFAs in TAG did not vary significantly (Tukey HSD, *p* > 0.05) between light treatments. An exception was evident for 16:1n-7, the amount of which was 2.3-fold lower in the HL than in the ML samples (Tukey HSD, *p* < 0.05). In contrast, the contents of 18:3n-3, 18:4n-3, and 20:5n-3, showed significant increases (by 1.7-fold, 1.4-fold, and 1.5-fold, respectively, Tukey HSD, *p* < 0.05) under HL. The contents of 18:2n-6 and 20:4n-6 were higher (by 1.7-fold and 1.9-fold, respectively) in the HL than in the ML samples (Tukey HSD, *p* < 0.05). The contents of other fatty acids did not show significant differences (Tukey HSD, *p* > 0.05) between the light treatments.

### 2.3. Chloroplast Ultrastructure

The tested chloroplast characteristics of *U. pinnatifida* were significantly affected by irradiance intensity after the 35-day exposure period (ANOVA, *p* < 0.05; Table 2). The average cross sectional area of chloroplasts was 1.6-fold lower in HL algae compared to ML algae (Turkey HSD, *p* < 0.05), indicating a net reduction in chloroplast size under HL exposure. The thylakoid stack number in chloroplasts was also lower (by 1.6-fold) in HL than in ML samples (Turkey HSD, *p* < 0.001). The distance between thylakoid stacks was slightly but significantly higher in HL samples than in the ML treatment group (Turkey HSD, *p* < 0.05), reflecting the decrease in thylakoid packing density at HL. The thylakoid membrane concentration had significantly lower values in HL samples (Turkey HSD, *p* < 0.05).

### 2.4. Photosynthetic Performance and Lipid Peroxidation

Long-term HL exposure induced a significant decrease in the photosynthetic performance of *U. pinnatifida*, where the maximum photochemical efficiency values of PSII, *F*_v_/*F*_m_, and the maximum rate of net photosynthesis, *P*_max_, were reduced by 45% and 48%, respectively, relative to those of the ML treatment group (Tukey HSD, *p* < 0.05; Table 3). Furthermore, HL algae exhibited significantly lower (by a factor of 2.3) levels of Chl (*a* + *c*) and a higher (by a factor of 1.4) threshold for irradiance required to saturate photosynthesis (*E*_k_) in comparison to ML algae (Tukey HSD, *p* < 0.05; Table 3), reflecting the reduced Chl antenna size in thylakoids under HL. Meanwhile, there were no significant differences in the rate of dark respiration (*R*_d_) between the HL and ML treatments (Tukey HSD, *p* > 0.05; Table 3).

The level of lipid peroxidation measured by malondialdehyde content (LPX–MDA) was assayed in order to explore the oxidative response of the algae to long-term light exposure. HL treatment resulted in significantly higher (by the factor 1.5) MDA content compared to the ML treatment group (Tukey HSD, *p* < 0.05; Table 3).

## 3. Discussion

It has been recognized that the changes in lipid composition of algae caused by environmental conditions reflect either modifications in the properties of cellular membranes or alterations in the relative rates of their production and utilization of storage lipids [28]. It is important to adjust the biochemical and physical properties of cell membranes to changing environmental conditions [6]. Lipids of thylakoid membranes, which are the site of photosynthesis and photoprotection, are particularly sensitive to irradiance intensity [8].

Glycolipids, monogalactosyldiacylglycerol (MGDG), digalactosyldiacylglycerol (DGDG), and sulfoquinovosyldiacylglycerol (SQDG) constituted about 50% of the total lipids of *U. pinnatifida* (Figure 1), and together with phosphatidylglycerol (PG), they are abundant chloroplast lipids representing the major building blocks for the thylakoid membrane assembly of brown algae [29] as well as another photoautotrophic organism [8]. Each of these lipids performs distinctive functions in photosynthetic membranes, where glycolipids, in addition to forming the lipid bilayer with photosynthetic complexes inserted into it, also participate in photosynthetic light reactions, chlorophyll biosynthesis, and the accumulation of light-harvesting proteins [8,30,31]. The inhibition of glycolipid biosynthesis probably influences the proliferation of thylakoids and chloroplasts and thus downregulates the processes of cell division and, consequently, the growth of brown algae [5].

Our results indicate that the character of changes in the major classes of thylakoid membrane lipids was specific under prolonged HL in *U. pinnatifida*. MGDG exhibited the most pronounced reduction compared to other chloroplast lipids; by contrast, DGDG, SQDG, and PG showed moderate declines (Figure 1). The response of these lipids to HL may result from their functional roles in keeping the proper structure and function of chloroplast membranes, as has been shown for thylakoids of higher plants [8,30,31].

MGDG, a non-bilayer-forming lipid, and DGDG, a bilayer-forming lipid, are two major components of thylakoid membranes in brown algae species [5]. Studies on higher plants have confirmed the essential effects of MGDG and DGDG on the structure and biogenesis of chloroplast membranes [11,32], pointing out that a deficiency in these lipids results in a reduction in thylakoids. Observations of *U. pinnatifida* chloroplasts have revealed significant reductions in thylakoid membrane concentrations, the number of thylakoid stacks, and the size of chloroplasts in HL algae (Table 2), which is a typical strategy for decreasing light absorption capacity [5]. Thus, the observed declines in the abundance of the MGDG and DGDG pools may contribute to the reduction in thylakoid membrane proliferation in brown macroalgae and thus prevent the absorption of excess irradiance. In addition, the interactions of MGDG and light-harvesting complex II (LHC II) are responsible for the stacking of the thylakoid membrane and facilitate the dense parking of proteins in the membrane [33]. In algae and higher plants, the suppression of light harvesting by PSII under HL benefits from the unstacking of grana thylakoids that is directed to the decrease in protein crowding [5,11]. In our study, the decrease in MGDG accompanied the reduction in the distance between thylakoid stacks (Table 2) and the decline in protein-packing density (PPD) in thylakoids that was evidenced by higher ratios of MGDG: Chl and DGDG: Chl (Figure 2) under HL. So, we can assume that such a reorganization of chloroplast structure firstly may ensure the dispersal of PSII and LHCII in membranes that help forward a low concentration of light-absorbing pigments (Table 3), thus decreasing the probability of capturing sunlight, and secondly guarantees the decrease in the intermolecular energy transfer between LHCII and PSII to avoid the overexcitation of photosystem. Furthermore, HL-induced thylakoid structural changes, such as the unstacking of thylakoids, are important for the repair of PSII and thereby have a key role in the adaptation of plants to high-light stress [34]. Taking into account that HL can induce the intensification of lipid peroxidation (LPX), as was observed in our experiments (Table 3), thylakoid unstacking also may facilitate the escape of reactive radicals from thylakoid membranes to avoid the chain reaction of LPX [6].

Aside from the essential role of MDGD and DGDG as constituents of photosynthetic membranes, they are also a part of photosynthetic complexes [11,35,36]. MGDG has been found in cyanobacteria photosystems and the cytochrome b_6_f complex, and DGDG was identified in PSII, PSI, and the light-harvesting complex of PSII (LHCII), as revealed by structural analyses of crystallized protein complexes [36,37]. MGDG and DGDG molecules are strongly linked to proteins and participate in stabilizing protein confirmation in PSII in thylakoid membranes [10,38]. Recent findings indicate that MGDG in membrane lipid bilayers switches LHCII from a light-harvesting to a more energy-quenching mode that dissipates harvested light into heat [36], indicating the important photoprotective function of this lipid. In this context, a pronounced reduction in MGDG content in *U. pinnatifida* is likely linked to the observed decrease in the maximum photochemical efficiency of PSII (Table 3) and thus contributes to the downregulation of photosynthetic activity and productivity in response to HL.

The balance between the proportion of the non-bilayer-forming MGDG and bilayer-forming DGDG affects the permeability of thylakoid membranes and the stability of the membrane bilayer [9]. The conical MGDG molecule has a relatively small head group and two acyl groups, which are usually represented by polyunsaturated fatty acids (PUFAs). Such a structure tends to pack into so-called hexagonal structures, which disrupt the membrane bilayer. In contrast, DGDG has a larger head group that forms a lipid bilayer, even when PUFAs are attached to the glycerol backbone [38]. Fine-tuning of the MGDG-to-DGDG-ratio affects membrane organization and protein folding and insertion, which exerts an impact on photosynthetic performance [11]. In this study, the MGDG-to-DGDG ratio was slightly reduced under HL (from 2.1 to 1.8; Figure 2), indicating non-principal changes from hexagonal to lamellar structures under prolonged HL exposure [39]. A decrease in the MGDG-to-DGDG ratio was also reported in red and green algal species exposed to HL [20,21]. In addition, to prevent the formation of the hexagonal structures and the consequent membrane fusion under unfavorable environmental conditions, MGDG can be converted to DGDG in the bilayer chloroplast membranes of higher plants [16]. It seems that the relatively stable level of DGDG and minor declines in the MGDG-to-DGDG ratio in HL *U. pinnatifida* are attributed in part to the conversion of MGDG to DGDG and may minimize damage to the photosynthetic apparatus, ensuring the maintenance of the chlorophyll content and photosynthetic capability of algae even under suboptimal conditions of irradiance. For these reasons, the rearrangement of the MGDG pool observed in our study could indicate remarkable alterations, which take place in the thylakoid membrane of brown algae, possibly affecting the interaction between pigment and proteins complexes.

Concerning regulations of fatty acid profiles in MGDG and DGDG, HL induced a significant decrease in n-3 PUFAs, mainly 18:4n-3 and 20:5n-3, and an increase in 16:0 and 18:1n-9 (Figure 3). However, there is no consistency in the literature on how light conditions affect the fatty acid profiles of these lipid classes. For example, in the marine microalga *Nannochloropsis oceanica*, PUFAs decrease under HL [19], whereas in *N. gaditana*, HL triggers an opposite effect, showing the increase in 20:5n-3 in MGDG and DGDG and a 16:1 increase [40]. A similar trend of PUFA increases was observed in MGDG and DGDG of the green macroalga *Bryopsis plumosa* under HL exposure [22]. This discrepancy between previous studies and our data could be a result of differences between genera and variations in experimental approaches. Specifically, it depends on the duration of exposure to high light.

The two major anionic lipids of photosynthetic membranes, SQDG and PG, were also sensitive to HL, showing moderate declines in their contents in *U. pinnatifida* after long-term HL exposure (Figure 1). Similar results were reported for HL-acclimated green and red macroalgae [20,21,22] as well as for HL-exposed green microalgal species [19,41]. SQDG and PG are acidic lipids. This is probably important for maintaining the balance of negative charges in thylakoid membranes [5]. Although the role of SQDG in algae photosynthesis has not been clear up to now, the content of this lipid is shown to be critical for their growth and development [28]. In brown microalgae, SQDG forms special domains within thylakoid membranes, which are separated from the antenna complexes [5]. One of the suggested functions of SQDG is participation in the stabilization of ATP-synthase complexes, and thus in the mechanism of ATP generation [30]. In this context, the decreased SQDG content in HL *U. pinnatifida* is probably related to the reduction in its net photosynthetic rate (Table 3), since a low capacity for photosynthetic assimilation necessitates a low rate of ATP production.

In higher plants, PG is found in chloroplast as well as in extraplastidic membranes [42], which is presumably true for marine macrophytes [28]. The only phospholipid of thylakoid membranes is PG. Although this lipid is present in a relatively small amount, it plays a crucial role in the formation of the structure and function of photosynthetic complexes ([42] and references therein). In particular, PG has been found in the cytoplasmic side of PSII, PSI, and LHCII in oxygenic photosynthetic organisms ([15] and references therein). PG is involved in the LHCII trimerization process [43] and in the assembly of trimeric PSI in the cyanobacteria [44]. PG also promotes the dimerization of the monomeric PSII core complexes and photosynthetic transport of electrons [45]. It has been shown that the depletion of PG is essential for the proper insertion of D1 protein into PSII complexes [46]. It has also been suggested that PG plays a direct structural role in binding antenna pigments in LHCII [37]. Taken together, the observed moderate decline in PG in HL *U. pinnatifida* (Figure 1) is likely involved in decreasing chlorophylls (Table 3), attenuating linear electron transport by reducing the photochemical efficiency of PSII (Table 3) and possibly the functional activity of PSI, and enhancing D1 protein turnover in order to adjust photosynthetic processes to suboptimal irradiance conditions.

The fatty acid composition of SQDG in *U. pinnatifida* was relatively unchanged in response to HL (Figure 3). In contrast, we observed significant changes in the fatty acid composition of PG in HL algae, with a decrease in 16:1n-3tr and 18:3n-3 in favor of 18:1n-9, 18:2n-6, and 20:4n-6 (Figure 3). In thylakoids of higher plants, 16:1n-3trans of PG is involved in the trimerization of LHCII [47], which is necessary for the formation of grana stacks [11] and the enhancement of the PSII light-harvesting efficiency [42]. For green microalgae of the genus *Nannochloropsis*, the detected increase in the trans isomer of 16:1 in PG under short-term high-intensity light has been proposed as a photoprotective mechanism to stabilize the trimerization of the LHCII [19,40]. In the case of HL *U. pinnatifida*, the decline in the level of 16:1n-3trans coincides with the reduction in the number of thylakoid stacks in chloroplasts (Table 2; Figure 3) and downregulation of PSII efficiency (Table 3), which are necessary for maintaining membrane integrity and, consequently, the photosynthetic processes of brown algae under long-term suboptimal irradiance conditions.

The general pattern for all the studied thylakoid membrane lipids, but not for SQDG, is the remodeling of fatty acid composition towards increasing the percentages of saturated and monounsaturated acyl groups over PUFAs in response to long-term HL exposure (Figure 3). This tendency has also been mentioned in marine green and red macroalgae [13,20]. Lipids are unique in generating lipid radicals through lipid peroxidation (LPX), which usually takes place under unfavorable conditions, particularly at irradiances higher than the saturated growth threshold [6,48]. Since LPX depends on the availability of PUFAs in membrane lipids [6], the observed PUFA reductions in MGDG, DGDG, and PG of HL *U. pinnatifida* are apparently related to photoprotective mechanisms carried out at the lipid level.

On a parity with thylakoid membrane lipid remodeling, a moderate decline in the content of the main cell membrane lipids, phosphatidylcholine (PC) and phosphatidylethanolamine (PE), was observed in *U. pinnatifida* after exposure to HL, while the abundance of phosphatidylinositol (PI) remained stable (Figure 1). The reduction in the content of PC and PE in response to HL exposure has also been reported for some species of macro- and microalgae [19,20,21,40]. It is not excluded that the decrease in PC in HL *U. pinnatifida* may result from shifting carbon flux from membrane lipid biosynthesis to triacylglycerol (TAG) synthesis at extraplastidic membranes. A similar process has been suggested for unicellular marine alga *N. oceanica* [49]. PC is associated with plasma membrane proteins, affecting their folding and functions [50]. The depletion of PC in eukaryotic algae is frequently accompanied with defects in growth [28] and thus may partially explain the reduction in growth rates of *U. pinnatifida* cultivated under suboptimal irradiances [26].

Lipid analysis of *U. pinnatifida* showed that under exposure to HL, excess irradiance was utilized and accumulated as the main TAG energy reserve (Figure 2). The variation in the level of TAG between light treatments was the highest among the studied lipids. Thus, under HL conditions, the metabolism of *U. pinnatifida* turned from polar lipid formation to energy storage, and more energy was provided for the synthesis of TAG, since this metabolic pathway consumes more energy [28]. Increased TAG levels under HL conditions have also been shown for the marine brown microalga *Thalasiosira pseudonana* [51] and for green and red macroalgae [20,21]. HL intensity affects lipid biosynthesis of the marine microalga *N. gaditana*, inducing the accumulation of TAG, together with the upregulation of genes involved in their biosynthesis [40]. The transcriptomic and lipidomic analyses of a unicellular freshwater microalga *Haematococcus pluvialis* suggest that the accumulation of TAG under high light is attributed to upregulation of de novo fatty acid biosynthesis at the gene expression level, as well as to the elevation of TAG assembly [41].

TAG has been considered as a PUFA reserve under HL conditions [49]. In the present study, *U. pinnatifida* demonstrated pronounced increases in n-3 PUFA at the expense of 20:5n-3, 18:4n-3, 18:3n-3, and n-6 PUFA at the expense of 20:4n-6 and 18:2n-6 in TAG after HL exposure (Table 1). Thus, in brown macroalgae, the activation of PUFA production and their reservation in TAG at HL is also directed to further demands for unsaturated fatty acids that can be transferred into thylakoid lipids, such as MGDG, DGDG and PG.

## 4. Materials and Methods

Specimens of the brown macroalga *Undaria pinnatifida* (Harv.) Suringar (1873) (Laminariales, Phaeophyceae, Ochrophyta) were collected in Peter the Great Bay, Sea of Japan, Russia. The collection was performed at a depth of 1.5–2 m in early spring. At this period, the seawater temperature in the field was about 7–8 °C, and the photosynthetically active radiation near the surface of the algae did not exceed 300 µmol photons m^−2^ s^−1^ under fine clear skies. After collection, plants were transported to the laboratory, cleaned, and cut into fragments of about 6 cm^2^. The part of the blade close to the meristem zone was used. Colored markers were attached to each fragment for the identification of individual plants. Fragmented algae were kept in a thermoregulated aquarium at 8 ± 0.4 °C and a light intensity of 260–270 μmol photons m^−2^ s^−1^ for three days. Seawater in the aquarium was enriched with Provasoli and continuously aerated. Light was supplied by a metal halide (MH) lamp (MLBOC400CU, Mitsubishi/Osram, Tokyo, Japan). The light:dark photoperiod was 12:12 h. A transparent glass lid covered the aquarium in order to prevent water splashes on the lamp. A light meter (LI-COR LI-189, Biospherical Instruments Inc., San Diego, CA, USA) was used to estimate the intensity of light.

### 4.1. Experimental Protocol

Following 3 d laboratory incubation, algal fragments were introduced into two experimental light treatments. Each treatment consisted of 42 fragments, which were maintained in separate glass aquaria, containing 30 L of aerated seawater. The temperature of seawater was kept at 8 °C using automatic thermostats (±0.5 °C IC Thermostat EX-003, Tokyo, Japan). The irradiance intensity was set at a moderate light level of 270 µmol photons m^−2^ s^−1^ (ML) in the first treatment group and at a high light level of 400 µmol photons m^−2^ s^−1^ in the second treatment group. These irradiance intensities were set based on optimal growth (i.e., 240–300 µmol photons m^−2^ s^−1^) and near inhibitory irradiance values, which are close to the upper limit of tolerance for the species (i.e., 420–460 µmol photons m^−2^ s^−1^) [26]. ML lams, which are described above, were used as a light source. The photoperiod was 12 h light: 12 h dark. The seawater in the aquaria was changed every three days. The duration of experimental exposure was 35 days, and then analyses of lipids, chloroplast ultrastructure, lipid peroxidation, photosynthetic performance, and pigment content of algae were conducted.

### 4.2. Lipid Analysis

To obtain enough material of a sufficient weight for lipid and fatty acid measurements, per treatment, eighteen exposed fragments (each from individual specimens) were divided randomly into three groups (each containing six exposed fragments) and used as three replicate samples. Before lipid extraction, algae were heated for 2–3 min in boiling water, dried by filter paper, and ground to a power. Lipids were extracted by homogenization with chloroform/methanol (1:2 *v*/*v*) [52]. The residue was extracted twice with small portions of chloroform/methanol (2:1 *v*/*v*) mixture. Combined extracts were filtered and separated into layers by adding water and chloroform. The lower layer was collected and evaporated under reduced pressure. The total lipid content was determined gravimetrically. The extract was stored at −20 °C.

Polar lipids were separated by silica gel TLC in solvent systems for plants [20]: chloroform–acetone–methanol–acetic acid–water (100:40:20:20:8 *v*/*v*) for the first direction and acetone–benzene–acetic acid–water (200:30:3:10 *v*/*v*) for the second direction. Nonspecific visualization was achieved by spraying the TLC plate with 10% H_2_SO_4_ in methanol followed by heating at 180 °C. For lipid identification the following specific reagents were used: molybdate reagent for phospholipids, 0.5% ninhydrin in acetone for aminolipids, Dragendorff’s reagent for choline-containing compounds, and anthrone reagent for glycolipids. The content of individual glycolipids and phospholipids was estimated by GLC, using 15:0 as an internal standard. Coefficients of calculation were 1.47, 1.78, 1.59, and 1.2 for MGDG, DGDG, SQDG, and TAG, respectively, and 1.45 for phospholipids. In addition, in order to further investigate changes in the chloroplast, the lipid-to-chlorophyll ratio was calculated from the MGDG or DGDG contents and the chlorophyll contents.

Fatty acid methyl esters (FAMEs) were prepared according to Carreau and Dubacq [53] by transmethylation of lipid samples by adding 1% Na in methanol, followed by heating for 15 min at 50 °C and then by adding 5% HCl in methanol, followed by heating for 10 min at 50 °C. The obtained FAMEs were purified by preparative silica gel TLC developed in benzene.

Gas chromatography analysis of FAMEs was carried out on a GC-2010 chromatograph (Shimadzu, Kyoto, Japan) with a flame ionization detector. A Supelcowax 10 (Supelco, Bellefonte, PA, USA) capillary column (25 m × 0.25 mm i.d., film thickness 25 μm) was held at 210 °C. The injector and detector temperatures were 250 °C. Helium was used as the carrier gas. FAMEs were identified by comparison with authentic standards and using a table of equivalent chain-lengths. The structures of FAs were confirmed by GC–MS of their methyl esters. The GC–MS analysis of FAMEs was performed at 160 °C with a 2 °C/min ramp to 240 °C that was held for 20 min. Injector and detector temperatures were 250 °C. A GCMS-QP5050A instrument (Shimadzu, Kyoto, Japan) and a MDN-5S capillary column (30 m × 0.25 mm i.d.) (Supelco Bellefonte, PA, USA) were used.

### 4.3. Chloroplast Ultrastructure Studies

The fixation and examination of algal fragments for morphometry of chloroplasts by electron microscopy were carried out as described [24]. Three replicate fragments per treatment were analyzed.

### 4.4. Lipid Peroxidation Analysis

The level of lipid peroxidation (LPX) was estimated in four pre-weighted replicate fragments per treatment. LPX analysis was performed by measuring the content of malondialdehyde (MDA, one of the end products of lipid peroxidation), using thiobarbituric acid (TBA) and reacting substance contents as described in [54]. Briefly, freeze-dried samples were ground in liquid nitrogen and extracted with 5% (*w*/*v*) trichloroacetic acid. The mixture was centrifuged at 12,000× *g* for 15 min at 4 °C, and the supernatants were used for analysis. To each 0.5 mL aliquot of the supernatant, 0.5 mL of 0.6% (*v*/*v*) TBA was added. The mixture was heated at 90 °C for 45 min and then quickly cooled on ice. After centrifugation at 1800× *g* for 10 min, the absorbance of the supernatant was recorded at 532 nm and 600 nm. MDA contents were calculated using an extinction coefficient of 156 mM^−1^ cm^−1^ at 532 nm after subtraction of the absorbance at 600 nm and the absorbance at 532 nm of a blank without the addition of seaweed material, normalized to biomass (wet weight, ww) and expressed as nmol g^−1^ ww.

### 4.5. Photosynthetic Performance and Pigment Analyses

Chlorophyll *a* fluorescence and oxygen concentration evolution were estimated in five and four replicate fragments per treatment, respectively, at the end of experimental exposure. The measurements were conducted between 10 a.m. and 11 a.m. local time.

The maximum quantum efficiency, *F*_v_/*F*_m_, of the algal fragment was measured using a portable Diving-PAM chlorophyll fluorometer (Heinz Walz GmbH, Effeltrich, Germany) [55]. The period of pre-incubation in darkness was 15 min.

The rates of dark respiration and net photosynthesis, which represent the cell energy requirements and availability of chemical energy, were measured as O_2_ concentration evolution in a closed transparent chamber (22 mL) fitted with a magnetic stirrer and a Clark-type electrode (CellOx 325, WTW, Weilheim, Germany) connected to an oxygen meter (OXI 197S, WTW, Weilheim, Germany). The chamber was filled with filtered (0.45 μm) seawater and kept in the water baths at a constant temperature (8 °C). Light was supplied by a slide projector halogen lamp L36W/25. Fragments were incubated for 30 min in the dark in order to determine dark respiration, and then photosynthesis versus irradiance curves (*P*–*E* curve) were recorded by incubating algal fragments at different irradiance intensities for 5–15 min each. Irradiance intensity was varied between 10 and 770 μmol photons m^−2^ s^−1^ using neutral density filters. Measurements were made below 50% of air equilibration oxygen concentration to avoid the possible inhibitory effects of high oxygen tensions [56]. Based on the obtained *P*–*E* curves, the following parameters were calculated: photosynthetic capacity expressed as maximum net photosynthetic rate (*P*_max_) and irradiance required to saturate photosynthesis (*E*_k_). The hyperbolic *P*–*E* curves were fitted according to the equation of [57], using KaleidaGraph version 3.0 (Synergy Software, Inc., Reading, PA, USA). Dark respiration and photosynthetic rates were normalized both to biomass (dry weight, dw) and expressed as mg O_2_ g^−1^ dw h^−1^.

Chlorophyll *a* and chlorophyll *c* pigments were estimated in four experimental fragments per treatment. The extraction procedure was conducted as described earlier [58]. A scanning UV-2100 spectrophotometer (Shimadzu, Tokyo, Japan) was used to measure the absorbance of the extracts. The chlorophyll concentrations were estimated using equations in [59]. Total Chl (*a* + *c*) contents were expressed as mg g^−1^ ww.

### 4.6. Statistical Analysis

To evaluate significant differences between light treatments, the data from different parameter measurements, obtained at the end of experimental exposure, were analyzed using ANOVA Turkey HSD test (*p* < 0.05). Analyses of homogeneity and normality of variances were performed using Levene’s test and Kolmogorov–Smirnov’s test, respectively. Data are presented as means and their standard errors.

## 5. Conclusions

In the case study of *Undaria pinnatifida*, it was shown for the first time how the functionality of thylakoid membranes is regulated in response to more light than is needed to saturate photosynthesis in brown macroalgae, which exhibit a wide range of light habitats and can experience long periods of high solar irradiation. A decrease in the proportions of thylakoid membrane lipids, MGDG, DGDG, SQDG, and PG, under high-light conditions indicates changes in the biogenesis of thylakoid membranes. Lipid metabolism switches from the formation of glycolipids and phospholipids to energy storage in the form of TAG. The remodeling of thylakoid membrane lipids is attended with a slowdown of photosystem II functioning, a reduction in photosynthetic capacity, and declines in chlorophyll content and protein packing density in thylakoid membranes. the observed remodeling of chloroplast membrane lipids and fatty acids, the ultrastructure of chloroplasts and photosynthetic performance augmented the ability of *U. pinnatifida* to counteract high-intensity light, thereby contributing to its survival potential under suboptimal irradiance conditions. The insights into the lipid remodeling of brown macroalgae under the influence of abiotic factors, especially the irradiance environment, are relevant to the commercial exploration of macroalgae for the production of lipid-based compounds, such as polyunsaturated fatty acids, as well as to the assessment of the resilience of these algal species.

## Figures and Tables

**Figure 1 plants-14-01818-f001:**
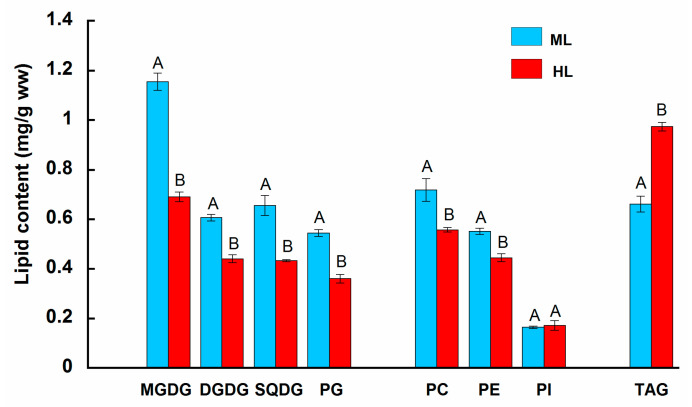
Content of the major lipid classes in *Undaria pinnatifida* after 35 days of exposure to moderate and high light intensities (ML, 270 µmol photons m^−2^ s^−1^ and HL, 400 µmol photons m^−2^ s^−1^, respectively). Means ± SE, *n* = 3. Bars of treatment groups marked with different letters are significantly different from one another (Turkey HSD, *p* < 0.05).

**Figure 2 plants-14-01818-f002:**
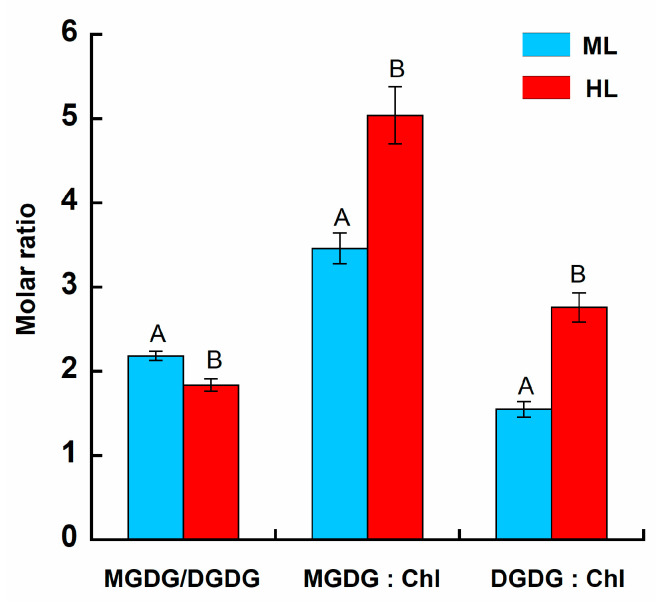
The ratios of MGDG-to-DGDG, MGDG-to-chlorophyll, and DGDG-to-chlorophyll in *Undaria pinnatifida* after 35 days of exposure to moderate and high light intensities (ML, 270 µmol photons m^−2^ s^−1^ and HL, 400 µmol photons m^−2^ s^−1^, respectively). Means ± SE, *n* = 3. Bars of treatment groups marked with different letters are significantly different from one another (Turkey HSD, *p* < 0.05).

**Figure 3 plants-14-01818-f003:**
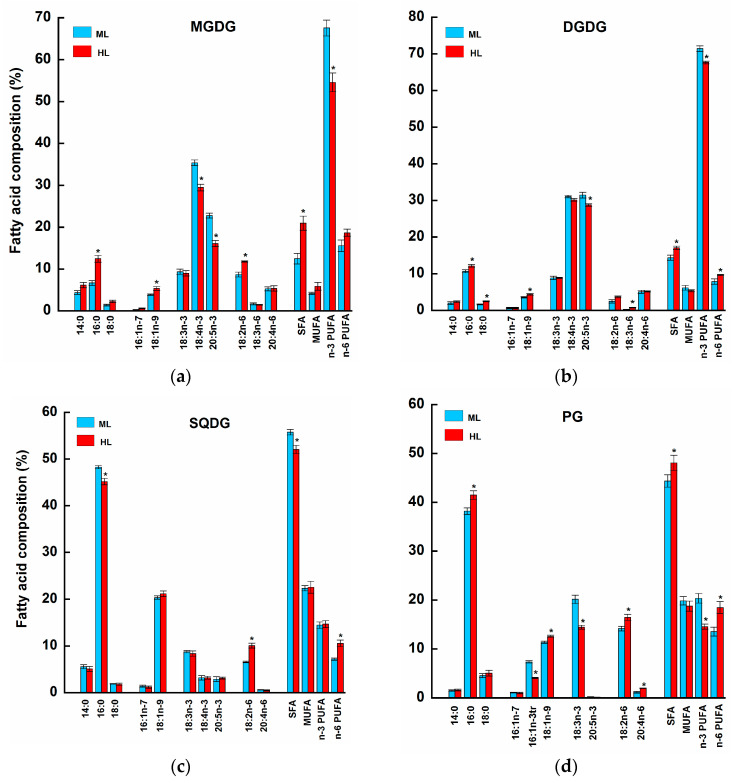
Fatty acid composition (%) of thylakoid membrane lipids (**a**) MGDG, monogalactosyldiacylglycerol; (**b**) DGDG, digalactosyldiacylglycerol; (**c**) SQDG, sulfoquinovosyldiacylglycerol; and (**d**) PG, phosphatidylglycerol in *Undaria pinnatifida* after 35 days of exposure to moderate and high light intensities (ML, 270 µmol photons m^−2^ s^−1^ and HL, 400 µmol photons m^−2^ s^−1^, respectively). Means ± SE, *n* = 3. Asterisks represent values significantly different from ML-exposed algae (Turkey’s HSD, *p* < 0.05).

**Table 1 plants-14-01818-t001:** Composition of fatty acids (%) in cell membrane lipids and triacylglycerols in *Undaria pinnatifida* after 35 days of exposure to moderate and high light intensities (ML, 270 µmol photons m^−2^ s^−1^ and HL, 400 µmol photons m^−2^ s^−1^, respectively).

Fatty Acid	PC		PE		PI		TAG	
	ML	HL	ML	HL	ML	HL	ML	HL
14:0	6.7 ± 0.4	8.9 ± 0.4	2.4 ± 0.4	2.2 ± 0.5	4.4 ± 0.2	3.4 ± 0.8	7.5 ± 1.0	7.5 ± 0.8
16:0	24.6 ± 1.2	27.8 ± 1.3 *	15.0 ± 1.1	16.0 ± 1.0	56.3 ± 1.5	52.1 ± 1.0	34.8 ± 1.4	24.1 ± 1.8 *
16:1n-7	0.4 ± 0.2	0.6 ± 0.1	0.8 ± 0.1	0.6 ± 0.2	1.7 ± 0.2	1.9 ± 0.8	3.4 ± 0.5	1.5 ± 0.6 *
18:0	3.8 ± 0.5	3.6 ± 0.4	3.6 ± 0.8	3.5 ± 0.3	5.7 ± 0.7	6.7 ± 1.0	12.0 ± 1.1	7.1 ± 1.1 *
18:1n-9	2.0 ± 0.7	2.0 ± 0.3	1.9 ± 0.3	2.9 ± 0.5 *	20.4 ± 1.1	21.7 ± 1.1	11.8 ± 1.8	11.3 ± 1.0
18:1n-7	–	–	0.5 ± 0.2	0.6 ± 0.2	–	–	0.8 ± 0.7	0.2 ± 0.1
18:2n-6	9.4 ± 0.6	8.1 ± 0.8	2.3 ± 0.5	2.1 ± 0.4	8.0 ± 0.8	10.1 ± 1.1	7.9 ± 0.5	13.3 ± 1.2 *
18:3n-6	0.4 ± 0.1	0.2 ± 0.2	–	–	–	–	0.2 ± 0.1	0.5 ± 0.5
18:3n-3	2.2 ± 0.1	2.4 ± 0.6	0.6 ± 0.1	0.4 ± 0.2	2.0 ± 0.8	2.1 ± 0.5	3.7 ± 0.5	6.2 ± 1.0 *
18:4n-3	0.9 ± 0.2	0.3 ± 0.2	0.6 ± 0.2	0.5 ± 0.1	–	–	5.1 ± 0.8	6.9 ± 0.8 *
20:3n-6	5.9 ± 0.7	3.7 ± 0.8 *	0.5 ± 0.2	0.7 ± 0.3	–	–	–	–
20:4n-6	34.5 ± 1.5	32.5 ± 1.7	59.5 ± 1.2	62.1 ± 1.1 *	1.4 ± 0.4	2.0 ± 0.4	5.8 ± 1.1	10.9 ± 1.4 *
20:5n-3	9.4 ± 1.1	10.0 ± 1.4	12.2 ± 1.4	8.5 ± 0.5 *	–	–	7.2 ± 1.1	10.6 ± 1.2 *
SFA	35.0 ± 2.1	40.3 ± 2.4	21.0 ± 2.3	21.7 ± 1.7	66.4 ± 2.4	62.2 ± 2.8	54.3 ± 3.5	38.7 ± 3.7 *
MUFA	2.4 ± 0.9	2.6 ± 0.5	3.2 ± 0.6	4.1 ± 0.9	22.1 ± 1.3	23.6 ± 1.9	16.0 ± 3.0	13.0 ± 1.9
PUFA n-3	12.4 ± 1.3	12.8 ± 2.3	13.5 ± 1.7	9.3 ± 0.8 *	2.0 ± 0.8	2.1 ± 0.5	16.1 ± 2.9	23.7 ± 3.0 *
PUFA n-6	50.1 ± 3.0	44.4 ± 3.5	62.3 ± 2.0	64.9 ± 1.9	9.4 ± 1.4	12.0 ± 1.5	13.7 ± 2.1	24.6 ± 3.2 *

Means ± SE, *n* = 3. Asterisks represent values significantly different from ML-exposed algae (Turkey’s HSD, *p* < 0.05). Abbreviations: PC—phosphatidylcholine; PE—phosphatidylethanolamine; PI—phosphatidylinositol; TAG—triacylglycerol; SFA—saturated fatty acid; MUFA—monounsaturated fatty acid; PUFA—polyunsaturated fatty acid; “–”—not detected.

**Table 2 plants-14-01818-t002:** Chloroplast morphometry in *Undaria pinnatifida* after 35 days of exposure to moderate and high light intensities (ML, 270 µmol photons m^−2^ s^−1^ and HL, 400 µmol photons m^−2^ s^−1^, respectively).

	Light Treatment	
Parameters	ML	HL
Area of chloroplast cross section, [µm^2^] ^a^	7.72 ± 0.05 ^A^	4.73 ± 0.17 ^B^
Thylakoid stack, number per chloroplast cross section ^b^	6.55 ± 0.11 ^A^	4.07 ± 0.14 ^B^
Distance between thylakoid stacks, [nm] ^c^	63.24 ± 0.12 ^A^	66.70 ± 0.14 ^B^
Thylakoid membrane concentration, number per µm^2 b^	4.99 ± 0.07 ^A^	4.12 ± 0.12 ^B^

Mean ± SE. Treatment groups with different capital letters are significantly different from one another (Turkey HSD test, *p* < 0.05). ^a^ Measured on 300 representative chloroplasts from each treatment. ^b^ Measured on 90 chloroplasts from each treatment. ^c^ Measured on 60 chloroplasts from each treatment.

**Table 3 plants-14-01818-t003:** Physiological parameters of *Undaria pinnatifida* after 35 days of exposure to moderate and high light intensities (ML, 270 µmol photons m^−2^ s^−1^ and HL, 400 µmol photons m^−2^ s^−1^, respectively).

	Light Treatment	
Parameters	ML	HL
*F*_v_/*F*_m_ (rel. units)	0.602 ± 0.014 ^A^	0.336 ± 0.030 ^B^
*P*_max_ (mgO_2_ g^−1^ dw h^−1^)	10.074 ± 0.433 ^A^	5.221 ± 0.138 ^B^
*R*_d_ (mg O_2_ g^−1^ dw h^−1^)	−3.282 ± 0.378 ^A^	−2.871 ± 0.279 ^A^
Chl (*a* + *c*) (mg g^−1^ ww)	0.409 ± 0.025 ^A^	0.177 ± 0.013 ^B^
*E*_k_ (µmol photons m^−2^ s^−1^)	177.5 ± 14.1 ^A^	251.6 ± 8.7 ^B^
LPX–MDA (nmol g^−1^ ww)	9.042 ± 0.560 ^A^	13.560 ± 0.551 ^B^

Means ± SE, *n* = 4–5. Treatment groups with different letters are significantly different from one another (Tukey HSD, *p* < 0.05). Abbreviations: *F*_v_/*F*_m_, maximum photochemical efficiency of PSII; *P*_max_, maximum rate of net photosynthesis; *R*_d_, dark respiration; Chl (*a* + *c*), chlorophyll (*a* + *c*) content; *E*_k_, minimum irradiance to saturate photosynthesis; LPX–MDA, lipid peroxidation measured by malondialdehyde content.

## Data Availability

Data are contained within the article.

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
