# Peer review of "Modulations of Photosynthetic Membrane Lipids and Fatty Acids in Response to High Light in Brown Algae (Undaria pinnatifida)"

_plants, 2025, doi:10.3390/plants14121818_

Round 1

Reviewer 1 Report

Comments and Suggestions for Authors

This study elucidates the photoprotective mechanisms by which the brown alga Undaria pinnatifida adapts to long-term high-light (HL) stress through remodeling of photosynthetic membrane lipids, shifts in fatty acid composition, and reorganization of chloroplast ultrastructure. However, the current manuscript requires major revisions to address the following critical issues:

  1. Conclude the Introduction by briefly highlighting the practical implications of the study's findings and potential directions for future research.
  2. Why chosen 400 and 270 µmol photons as the light intensity for normal and HL stress conditions? Did the author test in the lab that what light intensity is the threshold for HL stress? The authors should clarify as different research groups seemingly define HL with different light intensities.
  3. A significance threshold of P < 0.05 should be consistently applied, and all descriptions of 'significant' or 'pronounced' effects associated with P > 0.05 should be removed.
  4. The chloroplast ultrastructure and thin-layer chromatography (TLC) separation results should be presented in the main text with corresponding figures.
  5. The discussion regarding the impact of MGDG depletion on PSII efficiency lacks direct molecular evidence. Additionally, the proposed upregulation of TAG synthesis via 'gene expression' (lines 414–417) requires validation through transcriptomic profiling or enzymatic activity assays, which are currently absent from the manuscript.
  6. The manuscript contains a few minor errors, such as in line 161, 468. The authors are recommended to go through the whole manuscript for improvements.
  7. The repetitive descriptions of fatty acid composition changes (Figure 3) across multiple paragraphs should be consolidated to improve conciseness and avoid redundancy.
  8. The opening statement of the Conclusion claims 'the first discovery in the brown alga Undaria pinnatifida...' (lines 551–552), but lacks robust justification for its novelty within the broader context of algal photobiology research.
  9. The references cited in the article are outdated. Please cite recent literature from the past 3-5 years.

Author Response

C.1.1. Comment:

This study elucidates the photoprotective mechanisms by which the brown alga Undaria pinnatifida adapts to long-term high-light (HL) stress through remodeling of photosynthetic membrane lipids, shifts in fatty acid composition, and reorganization of chloroplast ultrastructure.

R1.1. Response:

We would like to thank the Reviewer 1 for the attention to our work and for the comments.

C.1.2. Comment:

Conclude the Introduction by briefly highlighting the practical implications of the study's findings and potential directions for future research.

R.1.2. Response:

In the original version of the manuscript (Introduction, Lines 84-88), we briefly highlighted the ecological and practical implications of the macroalgae ability to alter lipid metabolism under environmental influences, particularly fluctuations in light conditions. Thus, the addition of similar information at the end of the Introduction section would be a kind of repetition. However, in response to the Reviewer’s comment, we have added more detailed information on the practical implications of the obtained results into the Conclusion section of the revised manuscript :

“The insights into the lipid remodeling of brown macroalgae under the influence of abiotic factors, especially irradiance environment, are relevant to commercial exploration of macroalgae for the production of lipid-based compounds, such as polyunsaturated fatty acids, as well as to the assessment of the resilience of these algal species.”

C.1.3. Comment:

Why chosen 400 and 270 µmol photons as the light intensity for normal and HL stress conditions? Did the author test in the lab that what light intensity is the threshold for HL stress? The authors should clarify as different research groups seemingly define HL with different light intensities.

R.1.3. Response:

Probably the Reviewer missed the information regarding the bases for the chosen light intensities in our experiments. This is clearly explained in the Material and Methods section (4.2 Experimental protocol) of the original manuscript:

“These irradiance levels were set based on optimal growth (i.e. 240-300 µmol photons m-2 s-1) and near inhibitory irradiance values, which are close to the upper limit of tolerance for the species (i.e. 420-460 µmol photons m-2 s-1 ) [27]”.

For Undaria pinnatifida from Peter the Great Bay of the Sea of Japan, the irradiance levels for the optimal growth and for the HL inhibitory threshold were tested previously, and this information has been published (Skriptsova, 2008). So, our choice of the irradiance intensities for experimental design was based on the previous tests.

  1. Skriptsova, A.V. Biology and ecology of Undaria pinnatifida (Phaeophyta) in Peter the Great Bay of the Sea of Japan. In The current state of Aquatic Biological Resources, Proceedings of the Conference to the 70th Anniversary of the S.M. Konovalov, TINRO-Center: Vladivostok, Russia; 2008; 254–258.

C.1.4. Comment:

A significance threshold of P < 0.05 should be consistently applied, and all descriptions of 'significant' or 'pronounced' effects associated with P > 0.05 should be removed.

R.1.4. Response:

We are grateful to the Reviewer for the comment. In the revised manuscript, the misprints (P > 0.05 instead of P < 0.05) in the text of the Results section have been corrected consistently.

C.1.5. Comment:

The chloroplast ultrastructure and thin-layer chromatography (TLC) separation results should be presented in the main text with corresponding figures.

R.1.5. Response:

The TLC separation of polar lipids was carried out using generally accepted methods and solvent systems, which are described in details in the Material and Methods section, 4.3 Lipid analysis. This analysis is a routine procedure that does not require any additional evidence. Thus, presentation the results of TLC separation are redundant.

The transmission electron microscopy images we have are good enough to provide the measurements of morphometric characteristics of chloroplasts but the quality of the image’s resolution is not so high that it could be accepted for publication. We did not intend to provide the manuscript with the illustration of Undaria chloroplast ultrastructure because it is typical for representatives of brown algae (Solymosi 2012), and particularly is available for representatives of Undaria genus (Ginsburger-Voge et al 1992, Li et al 2020).

References cited above

Solymosi, 2012. Algae. Curr. Chem. Biol. 6:167–186;

Ginsburger-Vogel,  et al. 1992. Aquacul. 106, 171–181.

Li et al 2020 Marine Pollution Bulletin 153, 110978;

C.1.6. Comment:

The discussion regarding the impact of MGDG depletion on PSII efficiency lacks direct molecular evidence. Additionally, the proposed upregulation of TAG synthesis via 'gene expression' (lines 414–417) requires validation through transcriptomic profiling or enzymatic activity assays, which are currently absent from the manuscript.

R.1.6. Response:

Indeed, there is no direct molecular evidence for the impact of MGDG depletion on PSII efficiency and this topic needs further investigations. Meanwhile, we do not strictly state on it but make an assumption based on previously observed linkage between MGDG and structural/functional properties of photosystems in higher plants and cyanobacteria.

“… a pronounced reduction in MGDG content in U. pinnatifida is likely, linked to the observed decrease in maximum photochemical efficiency of PSII…”

As for the comment regarding TAG, we have clarified the sentence to make clear that upregulation of TAG synthesis via 'gene expression' has been shown previously for unicellular freshwater microalga Haematococcus pluvialis :

“The transcriptomic and lipidomic analyses of a unicellular freshwater microalga Haematococcus pluvialis suggests that accumulation of TAG under high light is attributed to up-regulation of de novo fatty acid biosynthesis at the gene expression level as well as to elevation of the TAG assembly pathways [42].”

C.1.7. Comment:

The manuscript contains a few minor errors, such as in line 161, 468. The authors are recommended to go through the whole manuscript for improvements.

R.1.7. Response:

Language has been edited by native English speaker.

C.1.8. Comment:

The repetitive descriptions of fatty acid composition changes (Figure 3) across multiple paragraphs should be consolidated to improve conciseness and avoid redundancy.

R.1.8. Response:

The purpose as well as the originality of the present study lies in the investigation of fatty acid composition changes within individual lipid classes of brown macroalgae in response to long-term high light exposure, that allowed identification of which specific membranes (e.g., thylakoid membranes) are primarily affected. Thus, it is important to show how fatty acid composition of particular lipid class responds to light changes. The description of the results on fatty acid profiles has been already consolidated into three blocks (paragraphs) according to the functional properties of the lipids – thylakoid membrane lipids (MGDG, DGDG, PG, SQDG); storage lipids (TAG) and cell membrane lipids (PC, PE, PI). This is a reason of the “repetitive” descriptions, which cannot be avoided more because it would be detrimental for the presentation of the obtained data.

C.1.9. Comment:

The opening statement of the Conclusion claims 'the first discovery in the brown alga Undaria pinnatifida...' (lines 551–552), but lacks robust justification for its novelty within the broader context of algal photobiology research.

R.1.9. Response:

The novelty of the obtained data within the broader context has been clarified.

C.1.10. Comment:

The references cited in the article are outdated. Please cite recent literature from the past 3-5 years.

R.1.10. Response:

Out of 60 references cited in the present manuscript, 15 articles are published in the period 2020-2025. This is all the recent literature on algae and higher plant lipid responses to high light that is available at present. Furthermore, a significant part of the articles, cited in our manuscript, represents "Classical" studies in the field of lipid biochemistry. We also believe that, for example, the article on transcriptomic and lipidomic analyses of microalgae, published in 2014, cannot be characterized as outdated.

Reviewer 2 Report

Comments and Suggestions for Authors

Peer Review Report on manuscript plants-3641016 “Modulations of photosynthetic membrane lipids and fatty acids in response to high light in brown algae (Undaria pinnatifida)”.

  1. Originality of Scientific Content, Novelty, and Accuracy

    Strengths:

  • The study presents original findings on how thylakoid membrane lipids and fatty acid metabolism are remodeled in the brown macroalga Undaria pinnatifida under high light (HL) stress, a topic that has received limited attention in this species.
  • The research integrates lipidomics, fatty acid profiling, and physiological observations (e.g., photosynthetic performance, chlorophyll content), which supports a comprehensive and accurate interpretation of lipid remodeling in response to HL exposure.
  • The focus on TAG as a PUFA reservoir and the shift in lipid metabolic pathways under HL are especially novel and add significant value to our understanding of photoacclimation in brown macroalgae.
  • The paper accurately cites and integrates prior findings on green, red, and marine microalgae, providing context and cross-species relevance.

      Minor concerns:

  • While novel for U. pinnatifida, some physiological observations (e.g., TAG accumulation under HL, PC depletion, lipid peroxidation susceptibility) are somewhat expected based on data from other algae. The degree of novelty could be better emphasized by elaborating on species-specific differences or potential applications.
  1. Significance of Content and Scientific Soundness

    Strengths:

  • The work is scientifically sound with a solid experimental foundation, including a detailed lipid analysis under different light treatments.
  • The study provides insight into how U. pinnatifida maintains homeostasis under HL by adjusting lipid composition, chloroplast structure, and energy storage—this has important ecological and biotechnological implications.
  • The metabolic shift from membrane lipid production to energy storage via TAG is clearly supported by data and gene expression trends.

     Significance:

  • The study contributes to broader understanding of photoprotection and stress adaptation in marine macroalgae, supporting both basic physiological knowledge and applied research (e.g., optimizing conditions for lipid-based bioproducts).
  • It may pave the way for algal strain selection or engineering aimed at biofuel or PUFA production.
  1. Quality of Presentation and Clarity of Language

    Strengths:

  • The structure of the manuscript is logical and coherent, with clearly divided sections for Introduction, Results, Discussion, and Conclusions.
  • Figures and tables (e.g., Figures 1–3, Tables 1–3) are referenced properly and meaningfully support the narrative.

     Language and clarity:

  • The scientific language is generally clear and professional, but there are some areas where sentence structure and terminology could be simplified or clarified to improve readability for a broader audience.

      Examples for improvement:

  • Replace overly complex constructions (e.g., “It is not excluded that the decrease in PC…” → “The decrease in PC may result from…”).
  • Improve transition phrases in discussion sections to clearly separate interpretation from speculation.
  • Define abbreviations (e.g., LPX, TAG, PUFA, PC, PE) at first use in each major section or remind the reader briefly if reused extensively.
  1. Interest to Readers

     Strengths:

  • This study will be of high interest to readers in marine biology, algal physiology, plant lipid metabolism, and biotechnology.
  • The emphasis on adaptive mechanisms of lipid remodeling under HL stress, and the functional shift toward energy storage, makes the article valuable for those interested in climate adaptation, aquaculture optimization, and bioenergy.

      Broader impact:

  • The insights into lipid remodeling in U. pinnatifida could be valuable for commercial applications, such as PUFA-enriched oils or stress-resilient algal strains.
  1. Suggestions for Improvement

    a) Clarify Mechanistic Hypotheses:

  • The manuscript would benefit from a more explicit explanation of the mechanisms linking light-induced stress, lipid peroxidation, and carbon flux redirection. A schematic model figure summarizing this would be very helpful.

      b) Strengthen Comparative Context:

  • Further emphasize how responses in U. pinnatifida differ or align with other algae (e.g., NannochloropsisThalassiosira), especially in PUFA dynamics and thylakoid remodeling.

      c) Language Polishing:

  • Minor editing of grammatical structures and some idiomatic expressions is recommended for smoother flow.

      d) Address Broader Ecological Implications:

  • In the Discussion or Conclusion, briefly speculate on the ecological role of these adaptations in natural environments (e.g., seasonal light fluctuations, tidal exposure).

Recommendation:

  • Accept with Minor Revisions.
    The manuscript presents significant and novel insights into light-dependent lipid remodeling in  pinnatifida. Minor improvements in language clarity, data presentation, and mechanistic discussion will elevate the impact and readability of this already strong work.
Comments on the Quality of English Language

Language and clarity:

  • The scientific language is generally clear and professional, but there are some areas where sentence structure and terminology could be simplified or clarified to improve readability for a broader audience.

Examples for improvement:

  • Replace overly complex constructions (e.g., “It is not excluded that the decrease in PC…” → “The decrease in PC may result from…”).
  • Improve transition phrases in discussion sections to clearly separate interpretation from speculation.
  • Define abbreviations (e.g., LPX, TAG, PUFA, PC, PE) at first use in each major section or remind the reader briefly if reused extensively.

Author Response

C.2.1. Comment:

The manuscript presents significant and novel insights into light-dependent lipid remodeling in  Undaria pinnatifida. Minor improvements in language clarity, data presentation, and mechanistic discussion will elevate the impact and readability of this already strong work.

R.2.1. Response:

We would like to thank the Reviewer 2 for the positive comments on the relevance and importance of our research.

C.2.2. Comment:

The scientific language is generally clear and professional, but there are some areas where sentence structure and terminology could be simplified or clarified to improve readability for a broader audience.

R.2.2. Response:

English was edited by a native speaker.

C.2.3. Comment:

  • Replace overly complex constructions (e.g., “It is not excluded that the decrease in PC…” → “The decrease in PC may result from…”).
  • Improve transition phrases in discussion sections to clearly separate interpretation from speculation.

R.2.3. Response:

Corrected and improved as suggested.

C.2.4. Comment:

Define abbreviations (e.g., LPX, TAG, PUFA, PC, PE) at first use in each major section or remind the reader briefly if reused extensively.

R.2.4. Response:

Abbreviations have been defined at first use in each major section of the manuscript, while the manuscript is provided with the list of all abbreviations at the end of the manuscript.

Reviewer 3 Report

Comments and Suggestions for Authors

The manuscript describes the changes in membrane compsition and structure of brown algae under high light conditions. The manuscript has been well-written and the results have been presented well. However, I have got a few concerns.

  1. I do not understand the purpose of this study. The goal of the research has not been made clear. Two different conditions have been used - one a baseline and the other as a test condition. Under the varying operating conditions, the microbes have undergone significant physiological changes, manifested as different lipid profiles and structures in cell membrane. However, weren't changes expected? Not the quantum of changes but just the fact that high irradiance would alter membrane composition? So, that could not be the goal of this research. The goal should have been to study the quantum of changes of the kind of changes?
  2. That brings me to the main problem with the study that it relies on one single experimental condition to justify its finding. Why did the authors rely on only one condition? Would it not have been better (and results more reliable) if at least two different light conditions (apart from baseline) had been used? Could we have seen a trend establishing the regulation of lipid levels based on light intensity? How do we rely on the "science" of the effects based on the limited experimental evidence that the authors have obtained? 
  3. The authors have picked light intensity close to the inhibitory levels. However, two important effects at play under these conditions have not been discussed at all: photo inhibition (related effect of photo blaching etc) and self-shading. For example, it is not clear what was the biomass density at the end of the experiments? Would biomass concentratrion at that level could have caused sufficient self-shading and could it have led to an improvement in photosynthetic performance.     

Despite my concerns, I think the authors have done reasonably good work with the data they have got. They should make major changes in the manuscript to pitch the research in a different way and improve Discussion part significantly.  

Author Response

C.1.1. Comment:

The manuscript describes the changes in membrane composition and structure of brown algae under high light conditions. The manuscript has been well-written and the results have been presented well. R.1.1. Response:

We would like to thank Reviewer 3 for the attention to our work and for the comments.

C.1.1. Comment:

I do not understand the purpose of this study. The goal of the research has not been made clear. Two different conditions have been used - one a baseline and the other as a test condition. Under the varying operating conditions, the microbes have undergone significant physiological changes, manifested as different lipid profiles and structures in cell membrane. However, weren't changes expected? Not the quantum of changes but just the fact that high irradiance would alter membrane composition? So, that could not be the goal of this research. The goal should have been to study the quantum of changes of the kind of changes? That brings me to the main problem with the study that it relies on one single experimental condition to justify its finding. Why did the authors rely on only one condition? Would it not have been better (and results more reliable) if at least two different light conditions (apart from baseline) had been used? Could we have seen a trend establishing the regulation of lipid levels based on light intensity? How do we rely on the "science" of the effects based on the limited experimental evidence that the authors have obtained?

R.1.1. Response:

We do not accept that the goal of our research is not clear. We would like to clarify that the goal of our study was to test a specific hypothesis put forward on the basis of currently available information for higher plants and unicellular algae. Brown multicellular algae (macroalgae are not microbes) form a specific systematic group whose characteristics of chloroplast ultrastructure, pigment-protein complexes (the fucoxanthin chlorophyll a/c binding proteins), lipid profiles, life cycle and etc. differ from , for example, those of the red and the green algae or higher plants. There is a big gap in our knowledge regarding the remodeling of lipids and their fatty acids in brown macroalgae as an effective response strategy to counteract environmental fluctuations, particularly those of irradiance. The hypothesis testing does not require a variety of light treatments. Based on two light treatments, we clearly prove the regulation of lipid levels of brown macroalgae in response to long-term exposure to high irradiance, which is near the upper limit of species tolerance. We believe that the obtained results are straightforward and statistically significant. While our results are somewhat expectable, it is, in fact, the first experimental demonstration of the high light-induced modulation of the lipid class and their fatty acid profiles in brown macroalgae and, in particular, being accompanied by simultaneous assessment of the algal chloroplast morphometry and photophysiological performance.As for the limitations of the experimental approaches, the potential directions for future research are always open. Since, it may not be entirely appropriate to extrapolate directly the responses of algal fragments to ecological scenarios, future studies should be focused on fully controlled field experiments.  

C.1.2. Comment:

The authors have picked light intensity close to the inhibitory levels. However, two important effects at play under these conditions have not been discussed at all: photo inhibition (related effect of photo blaching etc) and self-shading. For example, it is not clear what was the biomass density at the end of the experiments? Would biomass concentration at that level could have caused sufficient self-shading and could it have led to an improvement in photosynthetic performance.

R.1.2. Response:

Our work was dedicated to the remodelings of photosynthetic membrane lipids and their fatty acids in brown macroalgae, and the discussion of the results was structured according to this main topic.

In our experimental approach, algal fragments of about 6 cm2 were used. The experimental aquariums were large enough to avoid sufficient self-shading of the samples during experiments.

C.1.3. Comment:

Despite my concerns, I think the authors have done reasonably good work with the data they have got. They should make major changes in the manuscript to pitch the research in a different way and improve Discussion part significantly.

R.1.3. Response:

We would like to thank the Reviewer 3 for the appreciation of our work.

Discussion has been improved according to the comments of the Reviewers.

Round 2

Reviewer 1 Report

Comments and Suggestions for Authors

The results of the chloroplast ultrastructure and thin-layer chromatography (TLC) separation have not been included in the main text. The author must provide these results to substantiate the reliability of the data.

Author Response

Comment:

The results of the chloroplast ultrastructure and thin-layer chromatography (TLC) separation have not been included in the main text. The author must provide these results to substantiate the reliability of the data.

Response:

We would like to thank Reviewer 3 for the attention to our work and for the comments.

The TLC separation of polar lipids was carried out using generally accepted methods and solvent systems, which are described in details in the Material and Methods section, 4.3 Lipid analysis. The technique to the separation of complex mixtures of polar lipids from plants has been initially developed in our laboratory [Vaskovsky V.E. and Khotimchenko S.V. HPTLC of polar lipids of algae and other plants. J. High Resol. Chromat. 1982, 5, 635-636]. This article contains the image of the two-dimensional chromatogram (TLC) of model lipid mixture of macroalgae, representing the separation of polar lipids. It has been widely recognized and is used by researchers all over the world. This analysis is a routine procedure that does not require any additional evidence. Thus, presentation of the results of TLC separation is redundant. Earlier, this technique was used for the study of low light effects on Undaria pinnatifida. [Zhukova, N.V.; Yakovleva, I.M. Low light acclimation strategy of the brown macroalga Undaria pinnatifida: Significance of lipid and fatty acid remodeling for photosynthetic competence. J Phycol. 2021, 57, 1792–1804. doi: 10.1111/jpy.13209].

The transmission electron microscopy images we have are good enough to provide the measurements of morphometric characteristics of chloroplasts but the quality of the image’s resolution is not so high that it could be accepted for publication. We did not intend to provide the manuscript with the illustration of Undaria chloroplast ultrastructure because it is typical for representatives of brown algae (Solymosi 2012), and particularly is available for representatives of Undaria genus (Ginsburger-Voge et al 1992, Li et al 2020).

References cited above

Solymosi, 2012. Algae. Curr. Chem. Biol. 6:167–186;

Ginsburger-Vogel,  et al. 1992. Aquacul. 106, 171–181.

Li et al 2020 Marine Pollution Bulletin 153, 110978;

Thus, the methodological details of TLC separation, as well as the images of Undaria chloroplast ultrastructure are well-known and repeatedly tested, are not worth focusing on in the article.

Reviewer 3 Report

Comments and Suggestions for Authors

I am still not convinced that the research brings any "new" or meaningful contribution to science or provides new insights into algal physiology. The hypothesis has been framed in a way that any results would be a validation. Assessing difference between two randomly selected conditions is a demonstration and not an insight. 

However, considering the time, effort and good presentation, the research can be considered for publication.

Author Response

Comment:

I am still not convinced that the research any "new" or meaningful contribution to science or provides new insights into algal physiology. The hypothesis has been framed in a way that any results would be a validation. Assessing difference between two randomly selected conditions is a demonstration and not an insight.

Response:

We would like to thank Reviewer 3 for the attention to our work and for the comments.

The light treatments were not selected randomly. In the section Materials and Methods, the clear explanation of the treatment conditions is presented. “The irradiance intensity was set at a moderate light level of 270 µmol photons m-2 s-1 (ML) in the first treatment group, and at a high light level of 400 µmol photons m-2 s-1 in the second treatment group. These irradiance intensities were set based on optimal growth (i.e. 240-300 µmol photons m-2 s-1) and near inhibitory irradiance values, which are close to the upper limit of tolerance for the species (i.e. 420-460 µmol photons m-2 s-1 ) [26].”

We do not agree that our study is unconvincing. Really, it is the first study to demonstrate the strategy of brown macroalgae to counteract high-intensity light from view point of lipid metabolism. The results obtained showed the mechanism of the adjustment of the thylakoid membrane functionality and flexibility of lipid metabolism in the brown macroalga Undaria pinnatifida after long-term exposure to high light. Lipid metabolism switches from the formation of glycolipids and phospholipids to energy storage in the form of TAG. The remodeling of thylakoid membrane lipids is attended with a slowdown of photosystem II functioning, reduction in the photosynthetic capacity, as well as the declines in chlorophyll content and protein packing density in thylakoid membranes. The observed remodeling in chloroplast membrane lipids and fatty acids, the ultrastructure of chloroplasts and photosynthetic performance augmented the ability of U. pinnatifida to counteract high-intensity light, thereby contributing to its survival potential under suboptimal irradiance conditions. Thus, we do not accept that our research does not provide new insights into algal physiology.